# Estimating Excess Mortality Due to Prostate, Breast, and Uterus Cancer during the COVID-19 Pandemic in Peru: A Time Series Analysis

**DOI:** 10.3390/ijerph20065156

**Published:** 2023-03-15

**Authors:** Max Carlos Ramírez-Soto, Mariangel Salazar-Peña, Javier Vargas-Herrera

**Affiliations:** 1Facultad de Ciencias de la Salud, Universidad Tecnológica del Peru, Lima 15046, Peru; mariangelsp1021@gmail.com; 2Unidad de Telemedicina, Facultad de Medicina, Universidad Nacional Mayor de San Marcos, Lima 15001, Peru; jvargash@unmsm.edu.pe

**Keywords:** prostate cancer, breast cancer, uterus cancer, mortality, excess deaths, excess mortality, Peru, COVID-19

## Abstract

During the COVID-19 pandemic, most of the deaths in Peru were related to COVID-19; however, cancer deaths have also been exacerbated in the first months of the pandemic. Despite this, excess mortalities of prostate, breast, and uterus cancer are not available by age group and region from January to December 2020. Therefore, we estimated the excess deaths and excess death rates (per 100,000 habitants) due to prostate, breast, and uterus cancer in 25 Peruvian regions. We did a time series analysis. Prostate, breast, and uterus cancer death data for 25 Peruvian regions were retrieved during the COVID-19 pandemic in 2020, as well as data for up to 3 years prior (2017–2019) from the Sistema Informatico Nacional de Defunciones at the Ministry of Health of Peru. Deaths in 2020 were defined as observed deaths. The expected deaths (in 2020) were estimated using the average deaths over 3 preceding years (2017, 2018 and 2019). Excess mortality was calculated as the difference between observed mortality and expected mortality in 2020. We estimated that the number of excess deaths and the excess death rates due to prostate, breast, and uterus cancer were 610 deaths (55%; 12.8 deaths per 100,000 men), 443 deaths (43%; 6 deaths per 100,000 women), and 154 deaths (25%; 2 deaths per 100,000 women), respectively. Excess numbers of deaths and excess death rates due to prostate and breast cancer increased with age. These excess deaths were higher in men aged ≥ 80 years (596 deaths (64%) and 150 deaths per 100,000 men) and women aged 70–79 years (229 deaths (58%) and 15 deaths per 100,000 women), respectively. In summary, during the COVID-19 pandemic, there were excess prostate and breast cancer mortalities in 2020 in Peru, while excess uterus cancer mortalities were low. Age-stratified excess death rates for prostate cancer and breast cancer were higher in men ≥ 80 years and in women ≥ 70 years, respectively.

## 1. Introduction

Burden disease and mortality varies by location and time because of multiple factors such as environment, lifestyle, human biology, and healthcare. This measurement is fundamental to health decision-making. A recent study estimated that between 1 January 2020 and 31 December 2021, there were excess deaths of 18.2 million people worldwide because of the COVID-19 pandemic [1]. The global rate of excess mortality was 120.3 deaths per 100,000 of the population, and the excess mortality rate exceeded 300 deaths per 100,000 of the population in 21 countries, including Peru [1]. Therefore, COVID-19 infection was, potentially, the main cause of mortality between 2020 and 2021 globally. In addition to COVID-19 deaths, social distancing, and pandemic restrictions, societal and behavioral changes associated with the pandemic overstretched health care systems; other country-specific factors may have increased non-COVID-19 death rates, such as deaths from chronic diseases, including cancer [2]. Therefore, these changes to mortality patterns of chronic diseases may affect the excess mortality of non-COVID-19 deaths.

During the first 30 weeks of the COVID-19 pandemic in England and Wales, there was significant geographic variation in excess mortality for respiratory causes, but not for cancer [3]. A few months later in 2020, a modelling study suggested that substantial increases in the numbers of avoidable cancer deaths in England were to be expected because of diagnostic delays due to the COVID-19 pandemic in the UK [4]. In Belgium, there were considerable excess cancer mortalities observed during the initial peak of COVID-19 [5]. For prostate cancer, a study in Sweden showed an excess mortality rate in 2020 compared to previous years in all men [6]. In the United States, a study suggested that initial COVID-19 pandemic-related disruptions to care would have a small long-term cumulative impact on breast cancer mortality [7]. Finally, in the UK, the premature breast cancer deaths resulting from diagnostic delays during the first wave of the COVID-19 pandemic resulted in significant economic losses [8]. Due to this, there have been multiple changes globally in the services provision of cancer care from the point of diagnosis and treatment. These changes may have directly or indirectly resulted in excess cancer deaths.

A national lockdown was introduced across Peru on 16 March 2020 to reduce the potential impact of the COVID-19 pandemic on health services and the population. The national lockdown and overstretched health care system have been associated with an increasing concern about the effect on other patient groups requiring time-critical access to health care services and a decrease in most non-COVID-19 health services, for example, health services for patients with chronic diseases such as cancer, with whom timely diagnoses and treatment are vital to ensure patients’ survival. Since March 2020, in six cancer centers, rescheduling of treatments and diagnostic procedures was reported, especially in metastatic patients, with an average reduction of 40% in weekly care and treatments [9]. Additionally, there was an increase in cancer mortality in an oncology reference center in 2020, compared with in 2019 [10]. The postponement of cancer care, the fear in people with suspected cancer of becoming infected and choosing not to go to health centers, and the suspension of cancer screening programs, including diagnostic examinations for cancer and treatment, are a set of factors that could have directly or indirect affected cancer progression and death during the COVID-19 pandemic. Prostate cancer, breast cancer, and leukemia were responsible for 50.43%, 33.62%, and 32.78% of the 928 (17.27%) cancer deaths in Peru reported by a study between 15 March and 30 June 2020 [11]. In this study, the authors filtered records of neoplastic diseases according to the tenth revision of WHO’s International Statistical Classification of Diseases (ICD-10) to select the cause of death [11]. However, the IRIS software is currently used for coding the multiple causes of death and for the selection of the underlying cause of death [12]. Analysis with this software is used to reduce the information bias on the cause of death. Despite these published findings, excess cancer mortalities (prostate, breast, and uterus) are not available by age group and region from January to December 2020. Therefore, it is important to estimate the impact of the COVID-19 pandemic on excess cancer mortalities, as there is not enough information to date.

The objective of this study was estimated the excess mortality due to prostate, breast, and uterus cancer during the COVID-19 pandemic in Peru, from 1 January to 31 December 2020.

## 2. Materials and Methods

### 2.1. Study Design

We did a time series analysis following the Strengthening the Reporting of Observational Studies in Epidemiology reporting guidelines (STROBE; Appendix A Table A1. Checklist STROBE) [13]. In order to estimate the excess deaths and excess deaths’ rate due to prostate, breast, and uterus cancer, we used population-based electronic death records in Peru from death records extracted from the Sistema Informático Nacional de Defunciones (SINADEF; in English, ‘National Information Technology System’) at the Ministry of Health of Peru [14]. We used this data source because of its extensive information on death records in Peru (data may be lacking from the handbook registry). Underlying causes of cancer (from prostate, breast and uterine) deaths were obtained from death certificates and by the use of ICD-10 [15]. Underlying cause of death is defined by the World Health Organization as the disease that initiated the series of events that led directly to death. This condition may be the main reason for, or a reason contributing to, the event of death.

### 2.2. Ethical Aspects

We used publicly available death data of the SINADEF in Peru. They published these data fully anonymously as part of routine surveillance. Therefore, the study was exempt from review by an ethics board.

### 2.3. Population Study

The SINADEF data only includes information by age and sex, the year, the basic cause of death, the International Classification of Diseases 10th Revision (ICD-10), and the geographic region. For prostate cancer, the study population contained male death ≥40 years of age (40–49, 50–59, 60–69, 70–79, and ≥80 years), registered from 1 January 2017 to 31 December 2020. For breast cancer and uterine cancer, the study population contained female deaths ≥30 years of age (30–39, 40–49, 50–59, 60–69, 70–79, and ≥80 years), registered from 1 January 2017 to 31 December 2020. We used data made available, disaggregated by age and region. We used deaths due to prostate, breast, and uterus cancer registers for 25 Peruvian regions and divided into the data into two groups: (1) deaths from January through to December 2020 and (2) the preceding 3 years (2017–2019). We selected the death records whose cause of death A, B, C, or D (any of the 4 causes) was a death from prostate, breast, or uterus cancer. This information was entered into in the IRIS software to determine the underlying cause of death [12,16].

### 2.4. Use of IRIS Software for Automatic Coding and Selection of the Underlying Cause of Death

The underlying cause of death was selected using the software IRIS. The software IRIS is a tool for the automatic selection of the cause of death [16]. This software was developed, maintained, and deployed by the IRIS Institute in Germany, in collaboration with various countries in Europe. IRIS is based on the WHO-recommended international model death certificate form [12,16]. IRIS encodes the causes of death and selects the basic cause of death, following the rules contained in volume 2 of the ICD-10 [14].

In this study, we used the IRIS V5.8.0 software for automatic coding of causes of death [17]. IRIS V5.8.0 has in its algorithm the rules for the identification of COVID-19 as a cause of death. We used a dictionary of medical terms that was prepared based on data from the SINADEF [14]. Information was initially analyzed in IRIS for automated batch coding. In this analysis, we recovered 70% of the records with basic cause of death. Next, the interactive mode was used for the selection of the remaining 30% of the death records, with a coder in charge who was trained in the selection process of the basic cause of death using ICD-10.

### 2.5. Statistical Analysis

We estimated the excess deaths and the excess death rates due to cancer according to cancer type, age, and region. Expected cancer deaths in 2020 were obtained from the average of deaths in the 3 years preceding the pandemic (2017–2019). Cancer deaths (from prostate, breast, and uterine) reported from 1 January to 31 December 2020 were defined as observed deaths in 2020. Excess cancer deaths were defined as the difference between the number of observed cancer deaths in 2020 (during the COVID-19 pandemic) and the number of expected cancer deaths based on past deaths in the previous 3 years (average number of cancer deaths that occurred between 2017 and 2019) [2,18]. Excess mortality also was calculated using the P indicator (P-score) [18], which is the quotient of this difference between observed and expected deaths divided by expected deaths multiplied by 100 (expressed in percent) ((observed deaths−expected deaths)/expected deaths × 100%) [14]. We calculated the cancer mortality rate observed and cancer mortality rate expected (per 100,000 habitants) by dividing the number of deaths per cancer type, region, and age by the estimated population of each region. The excess death rates by cancer type, age, and region per 100,000 population were calculated as the difference between the mortality rate observed and the mortality rate expected [18]. The population used to calculate the mortality rate was obtained at the Instituto Nacional de Estadística e Informática, Peru (INEI, in English ‘National Institute of Statistics and Informatics’) [19]. Odds ratio (ORs) with 95% confidence intervals (CIs) for the comparison between the observed deaths vs. expected deaths were estimated by age and cancer type. We summarized our results in a meta-analysis using the weights indicated. The Mantel–Haenszel method was used to calculate the random effects estimates, since there is no uniformity across reported deaths by age. We considered *p*-values < 0.05 as significant for meta-analytic methods. We performed this meta-analytic method in Review Manager Software (RevMan V.5.3 software (Cochrane)) [20].

## 3. Results

From 1 January 2017 to 31 December 2020, a total of 16,607 suspected prostate, breast, and uterus cancer deaths was identified. Of these deaths, we included 5087 prostate cancer, 4525 breast cancer, and 2601 uterus cancer deaths, after using the IRIS V5.8.0 software for automatic coding of causes of death, and selected the underlying cause of death (Figure 1). The monthly observed deaths distribution for prostate, breast, and uterus cancer between January 2017 and December 2020 showed an increase in deaths during the COVID-19 pandemic in 2020. Prostate cancer death was most prevalent among the three causes of death (Figure 2).

### 3.1. Excess Deaths and Excess Death Rates by Cancer Type

In 2020, there was a total of 1729 prostate cancer deaths (36.2 deaths per 100,000 men), 1463 breast cancer deaths (19.8 deaths per 100,000 women), and 766 uterus cancer deaths observed (10.3 deaths per 100,000 women). In comparison with the expected deaths (per 100,000) during the same period, the numbers of excess deaths and the excess death rate by prostate, breast, and uterus cancer were 610 (55%) prostate cancer deaths (12.8 deaths per 100,000 men), 443 (43%) breast cancer deaths (6 deaths per 100,000 women), and 154 (25%) uterus cancer deaths (2 deaths per 100,000 women) (Figure 3A,B).

The monthly estimates of excess mortality due to prostate, breast, and uterus cancer in Peru are shown in Figure 4. There were excess prostate cancer deaths from May to July 2020. We also found excess breast cancer deaths from April to September 2020. Although lower, there was an excess of uterus cancer deaths from October to December 2020 (Figure 4).

### 3.2. Excess Deaths and Excess Death Rates by Cancer Type and Age

Excess deaths (number and %) and excess death rates (per 100,000) by prostate cancer increased with age and were higher in men aged ≥ 80 years (380 deaths (64%) and 150 deaths per 100,000 men) (Figure 5A). Excess deaths (number and %) and excess death rates (per 100,000) by breast cancer also increased with age and were higher in women aged 70–79 years (84 deaths (58%) and 15 deaths per 100,000 women) (Figure 5B). Excess deaths (number and %) by uterus cancer varied with age, and excess death rates (per 100,000) increased slightly with age (Figure 5C).

### 3.3. ORs for Observed Deaths vs. Expected Deaths by Cancer Type and Age

Men and women had 1.55 (95% CI 1.43 to 1.67) and 1.43 (95% CI 1.32 to 1.55) times increased odds of observed deaths for prostate cancer and breast cancer compared with expected deaths, respectively (Figure 6). Men in the 50–59-year age group had 1.69 (95% CI 1.07 to 2.67) times increased odds of observed deaths by prostate cancer compared with expected deaths (Figure 6A). Women in the 70–79-year age group had 1.58 (95% CI 1.28 to 1.94) times increased odds of observed deaths by breast cancer compared with expected deaths (Figure 6B). There were not increased odds of observed deaths for uterus cancer compared with expected deaths (Figure 6C).

### 3.4. Excess Deaths and Excess Death Rates by Cancer Type and Region

The excess deaths (number and % (P-score)) and excess death rates (per 100,000) varied between regions (Figure 7). For prostate cancer, the number of excess deaths varied from zero deaths in Amazonas to 284 (76.6%) deaths in Lima, and excess death rates varied from zero deaths per 100,000 men to 130.3 deaths per 100,000 men. For breast cancer, the highest number of excess deaths and excess death rates (per 100,000 women) occurred in the Lima (382 deaths; 16.1 deaths per 100,000 women), Arequipa (23 deaths; 6.3 deaths per 100,000 women), and Pasco regions (18 deaths; 30.5 deaths per 100,000 women). For uterus cancer, the highest number of excess deaths and excess death rates (per 100,000 women) also occurred in the Lima (91 deaths; 3.8 deaths per 100,000 women), Arequipa (17 deaths; 4.7 deaths per 100,000 women), and Piura regions (14 deaths; 3.1 deaths per 100,000 women) (Figure 7).

## 4. Discussion

During the first months of the COVID-19 pandemic in 2020 in Peru, restrictions placed to enforce physical space distancing led to difficulty in accessing health services, including cancer centers. Therefore, in this study, we estimated the excess deaths and excess death rates due to prostate, breast, and uterus cancer during the COVID-19 pandemic in 25 Peruvian regions in 2020. In this study, we found that there was an excess of prostate and breast cancer mortalities, while excess uterus cancer mortalities were low.

### 4.1. Potential Explanations and Implications

Several studies demonstrated that prostate biopsies and diagnoses as well as monitoring and treatment rates decreased during the COVID-19 pandemic, particularly during the peak of the pandemic [21,22]. Results of a study in the United States demonstrated that prostate biopsy and PC diagnosis rates decreased from an estimated 97 cases to an estimated 573 cases during the COVID-19 pandemic, particularly during the peak of the COVID-19 pandemic [23]. Similar to these findings, a study in public cancer hospitals in Peru found that of the 98 patients who died: 73.5% died from cancer progression, 18.4% from COVID-19, and 8.1% from undetermined causes; advanced disease stage and discontinuation of therapy were risk factors. [24]. A cohort study in Belgium found an excess of 50 prostate cancer deaths and an expected excess of 105 deaths in April 2020 [5]. Another cohort study in England found excess 1-year mortalities in prostate cancer patients during the COVID-19 pandemic [25]. Similarly, since the lockdown was introduced across Peru in 2020, considering the overstretched health care system, rescheduling of treatments and diagnostic procedures in oncological patients occurred [10,24]. There was an increase in the mortality rate of prostate cancer compared to previous years because of this, and, hence, an excess of mortality from prostate cancer. Although some studies revealed an association in mortality between prostate cancer and COVID-19 [26], in our study, we included deaths whose main cause of death was prostate cancer, which was analyzed in IRIS. Similarly with our findings, a study in the United States found that non-COVID-19-related prostate cancer was more prevalent than in COVID-related cancer deaths during the first year of the COVID-19 pandemic (from March to December) [26,27]. On the other hand, according to the WHO, estimated crude mortality rate in 2020 for prostate cancer in males from aged 40+ was 41.4 deaths, while in our study we found a rate of 36.2 deaths [28]. Likewise, as per the WHO, our findings reveal that prostate cancer death rates in 2020 increased with age and were high in men 80 years of age or older [28,29]. Because mortality rates were high in 2020, compared with expected mortality rates in the same year, the excess mortality rate in men 80 years and older was 150 deaths per 100,000 men.

A systematic review found evidence of a moderate quality for diminished screening of both breast and cervical cancers in several countries during the COVID-19 pandemic [30]. Likewise, a modelling study in the United States predicted that initial COVID-19 pandemic-related disruptions in breast cancer care would have a small long-term impact on breast cancer mortality [7]. Another population-based modelling study from the UK predicted an increase in the number of deaths due to breast cancer up to year 5 after diagnosis (7.9–9.6%), corresponding to between 281 (95% CI 266–295) and 344 (329–358) additional deaths [4]. Before the COVID-19 pandemic (from 2003 to 2017), a study in Peru estimated a breast cancer mortality rate of 9.97 per 100,000 women years [31]. Similarly, in this study, we found an expected mortality rate of 13.8 per 100,000 women for 2020. This is probably due improvements in the coverage of deaths in Peru in recent years, as a result of the SINADEF. However, as in other studies, the restrictions imposed to prevent the transmission of COVID-19, and the collapse of the health system, have entailed multiple changes in the services provision of cancer care from the point of diagnosis and treatment, resulting in excess mortalities from breast cancer in Peru. These findings of excess mortalities from breast cancer over a year (443 excess deaths (43%)) are similar to a modeling study in Canada, where it was found that a six-month interruption could have led to an additional 250 cancer deaths during the COVID-19 pandemic [32]. Although we did not find estimates of excess mortality from breast cancer by age, it is important to note that excess mortality increased with age and was higher in people 70 years of age and older. Limited access to health services, fear of contagion, and other comorbidities likely contributed to this.

In our study, we found that excess mortality from uterus cancer was relatively low (25%; 154 deaths) compared with prostate or breast cancer. Several studies had reported a decrease in screening and diagnoses tests for uterine cancer during the pandemic. For example, a study in India estimated a 2.52% to 3.80% increase in the cervical cancer deaths with treatment restrictions, resulting in 18,159 to 53,626 life years [33]. The restrictions of the COVID-19 pandemic may have indirectly influenced screening, diagnoses, surgeries, and treatments of uterine cancer patients; changes in incidence and mortality due to the COVID-19 pandemic have also been projected from 2020 to 2044 in Australia [34]. Therefore, this could lead to excess uterus cancer deaths. The excess mortalities from uterus cancer also may be explained by other factors such as an increasing capacity for uterus cancer screening, resulting in an increase of new cancer cases and deaths. Other factors include the changes in uterus cancer risk factors and shifting female population age structures.

In our study, we found that the excess mortality from prostate and breast cancer was higher between May and September in 2020. Interestingly, these findings coincide with the peak mortality from COVID-19 and excess mortality from all causes in Peru [18,35]. This also coincides with several studies that reported excess cancer mortality in the first 6 months of the COVID-19 pandemic [4,5,6]. On the other hand, it is also important to highlight that excess mortality was predominant in the largest cities of Peru, such as Lima. This is because most of the Peruvian population lives in Lima. In addition, most of the highly complex health establishments and public and private cancer centers are in Lima.

### 4.2. Limitations

Our study has several limitations. First, the national system for registering death varies in quality. Although deaths are registered online at the SINADEF, deaths continue to be registered on paper in some locations of Peru where there is no access to the SINADEF. This could have prevented the correct reporting of cancer death causes in some Peruvian locations. Therefore, this can result in a possible bias for the estimation of excess cancer mortalities. Secondly, early in the COVID-19 pandemic in Peru, no SARS-CoV-2 molecular tests were available, and rapid immunochromatographic tests were used instead; therefore, due to the low sensitivity and specificity of these diagnostic tests, many deaths could have occurred by COVID-19-related cancer, but these could only have been considered as cancer deaths. It is noteworthy that death due to COVID-19 is not considered without a positive SARS-CoV-2 laboratory test in the Peruvian health system. Thirdly, we used a simple method to estimate excess mortality from prostate, breast, and uterine cancer. In contrast, studies that estimated excess cancer mortality used mathematics models to estimate expected mortality; therefore, making comparisons of excess cancer mortality between studies difficult. Fourthly, the magnitude and distribution of observed cancer deaths in 2020 could have changed because of behavioral factors and social or economic responses due to national lockdowns and other strict restrictions to prevent COVID-19 transmission; these may have affected the estimations of excess mortality from cancer. Another limitation was lack of data to confirm if the deaths were due to exacerbation of cancer symptoms or COVID-19 related comorbidity. Despite these limitations, the strength of our study is that the method used to select the underlying cause of death using the IRIS software allowed for the standardization of the procedure according to the WHO’s recommendations, reducing the information bias on the cause of death. In addition, our approach allows for international comparability in relation to the causes of death.

### 4.3. Implications for Public Health

Our findings on mortality excess have several implications for public health in Peru. First, estimates of national cancer mortality excesses provide essential information for policy makers, oncological health service planning, and public health policies. Secondly, these findings could serve as a basis for future cancer control interventions, as screening for prostate, breast and uterus cancer, chemotherapy treatments, and surgeries are crucial for long-term survival. Third, our findings also highlighted the need for policies that could potentially lessen the disruption of cancer diagnoses, treatment, and surgery in any future health emergencies. Additionally, they can be used in the prioritization of control programs and oncological health service planning for future health crises in Peru. Finally, our findings can be used to estimate the potential effects of COVID-19 restrictions on long-term prostate, breast, and uterus cancer mortality under different scenarios and plan the demand for oncological services over the coming years.

## 5. Conclusions

During the COVID-19 pandemic in 2020 in Peru, there was excess prostate and breast cancer mortalities, while excess uterus cancer mortalities were low. Age-stratified excess death rates for prostate cancer and breast cancer were higher in men ≥ 80 years and in women ≥ 70 years. These findings may be useful to understand how the COVID-19 pandemic indirectly affected cancer mortality, because of changes in the utilization of health services, the restrictions imposed by the government and the lack of accessibility to health services. In addition, excess mortalities provide a key tool in evaluating the effects of an ongoing pandemic.

## Figures and Tables

**Figure 1 ijerph-20-05156-f001:**
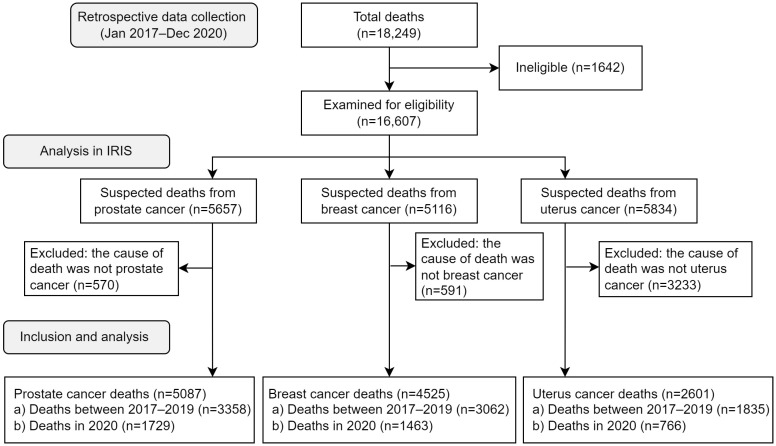
Study flow chart of excess cancer deaths using the STROBE reporting guidelines.

**Figure 2 ijerph-20-05156-f002:**
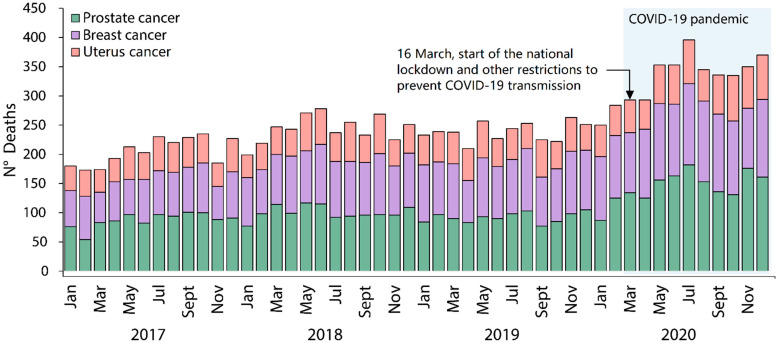
Trends in monthly observed deaths for prostate, breast, and uterus cancer between 1 January 2017 and 31 December 2020 in Peru.

**Figure 3 ijerph-20-05156-f003:**
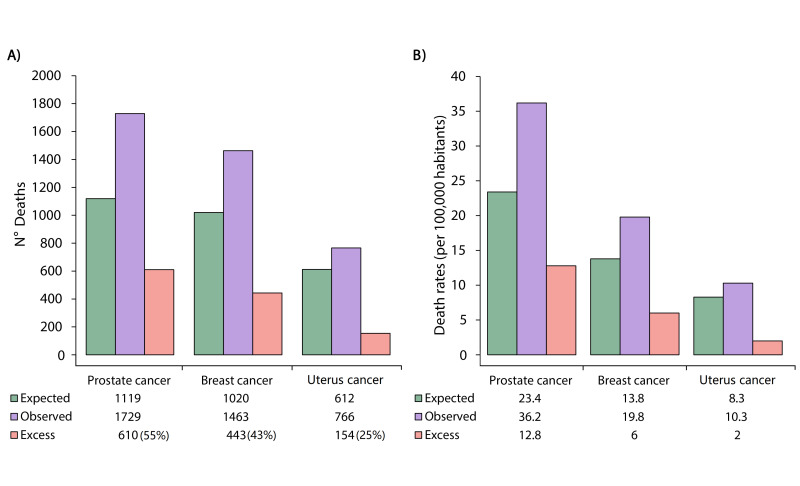
Deaths observed, deaths expected, and excess numbers of (**A**) deaths and (**B**) excess death rates for prostate cancer, breast cancer, and uterus cancer in Peru.

**Figure 4 ijerph-20-05156-f004:**
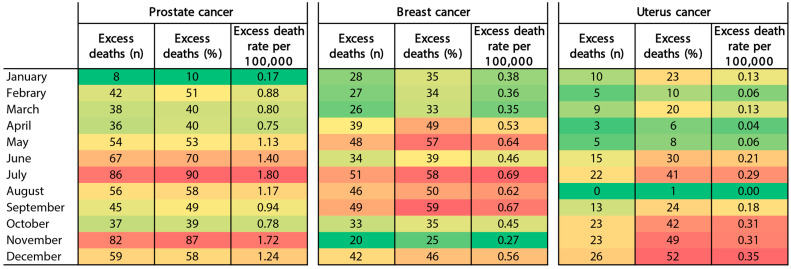
Trends in monthly excess deaths for prostate, breast, and uterus cancer between January and December 2020 in Peru. Color intensity is proportional to rank number (denoted by dark red for “highest excess” to “lowest excess” denoted by dark green).

**Figure 5 ijerph-20-05156-f005:**
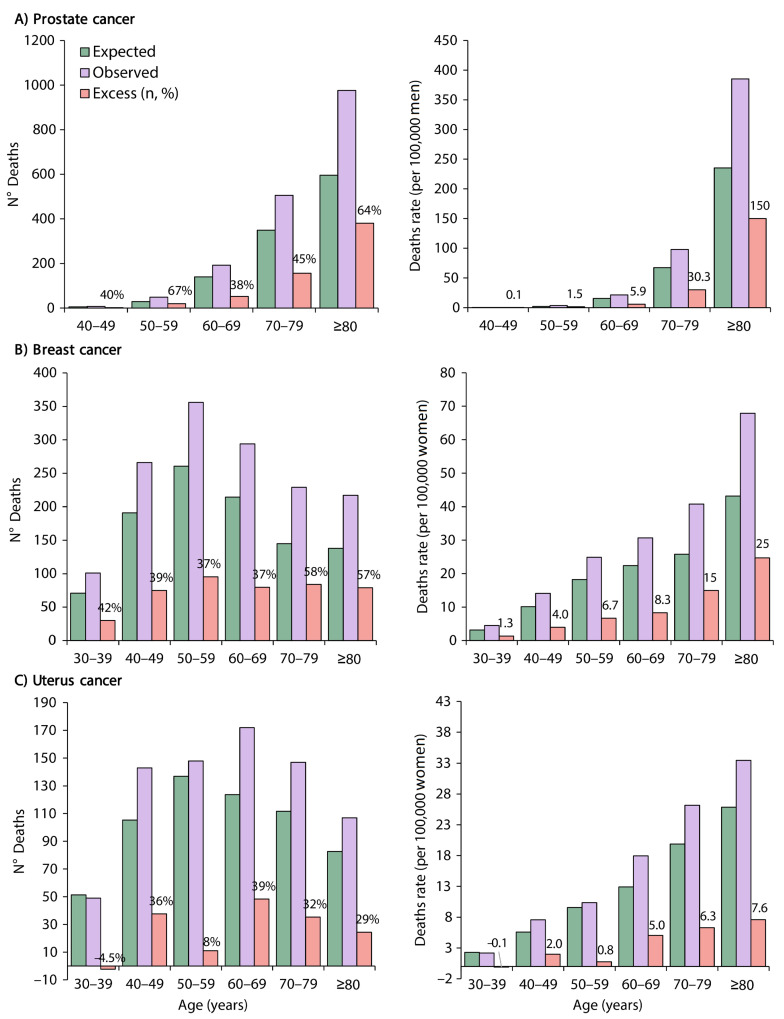
Deaths observed, deaths expected, and excess numbers of deaths and excess death rates by (**A**) prostate cancer, (**B**) breast cancer, and (**C**) uterus cancer, stratified by age in Peru.

**Figure 6 ijerph-20-05156-f006:**
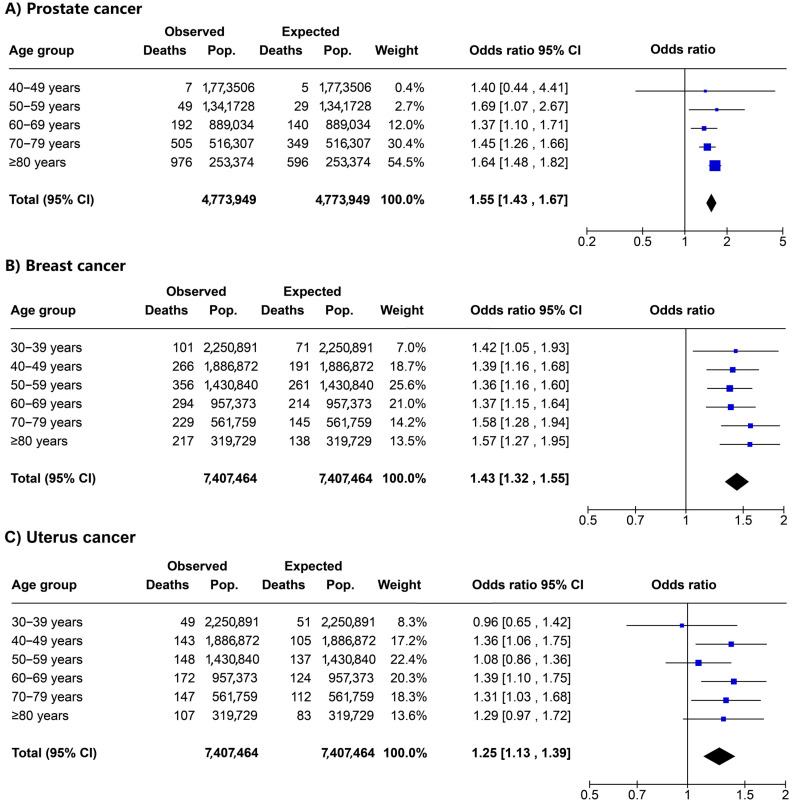
Odds ratio for excess deaths stratified by age for (**A**) prostate cancer, (**B**) breast cancer, and (**C**) uterus cancer in Peru. Observed deaths in 2020 vs. expected deaths in 2020.

**Figure 7 ijerph-20-05156-f007:**
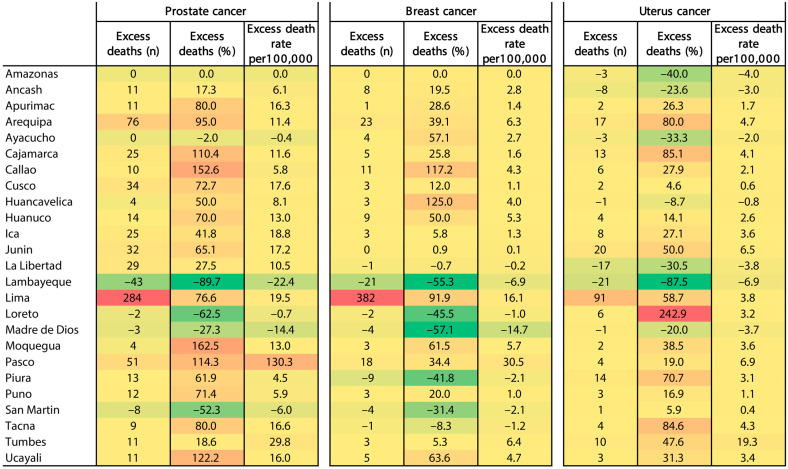
Excess deaths (n, % (P-score)) and excess death rates (per 100,000 habitants) by prostate cancer, breast cancer, and uterus cancer, stratified by regions in Peru. Color intensity is proportional to rank number (from 382 denoted by dark red, to −89.7 denoted by dark green). Negative values mean a reduction in deaths.

## Data Availability

The data in this study are publicly available at the National System of Deaths (SINADEF). Available online: https://www.datosabiertos.gob.pe/dataset/informaci%C3%B3n-de-fallecidos-del-sistema-inform%C3%A1tico-nacional-de-defunciones-sinadef-ministerio (accessed on 28 December 2021); INEI, Peruvian population: https://www.inei.gob.pe/estadisticas/indice-tematico/population-estimates-and-projections/ (accessed on 7 January 2022).

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
