# Peer review of "Estimating Excess Mortality Due to Prostate, Breast, and Uterus Cancer during the COVID-19 Pandemic in Peru: A Time Series Analysis"

_ijerph, 2023, doi:10.3390/ijerph20065156_

Round 1

Reviewer 1 Report

Journal of IJERPH (ISSN 1660-4601)

Manuscript ID: ijerph-2175734

Title 

Estimating Excess Mortality due to Prostate, Breast, and Uterus Cancer during the COVID-19 Pandemic in Peru: A Retrospective, Cross-Sectional Study

ABSTRACT

1.     The abstract missing the overall objective of the study’s authors didn’t clearly address the health issues caused by COVID-19 pandemic in the number of these cancers death and how they get the hypothesis of this study.

2.     English must be improved.

INTRODUCTION

1.     The first sentence of the manuscript “Burden disease and mortality varies by location and time.” Doesn’t have a scientific meaning authors must mention how and why because that’s due to multiple factors such as environment, lifestyle and genetics.

2.     Authors missing a reference concerning “there were excess deaths of 18,2 million people worldwide”

3.     This sentence is not clear and must be re-written “Therefore, COVID-19 infection was potentially a main cause of mortality. In addition to COVID-19 deaths, social distancing, and pandemic restrictions, societal and behavioral changes associated with the pandemic overstretched healthcare systems; other country-specific factors may have increased non- COVID-19 death rates, such as deaths from chronic diseases, including cancer [2]”

4.     Authors need to present a clear link between the increase in deaths by Covid19 not just because of the infection but also due to lifestyle changes which may affect the screening programs, diagnostic delays, and treatment response to author diseases including cancers.

5.     The English are too poor.

MATERIALS AND METHODS

1.     Design Study

a.     Usually, we say study design I don’t know if this title is suggested by the journal formatting.

b.     Authors must clarify the design of their study in this section not why they chose to use the population-based electronic death 82 records in Peru.

2.     Population Study 

a.     Ethical approval and IRB registration number must be added to this section.

b.     Usually, we present the date by month, day than the year: the date “1 January 2017” must be presented as January 1st, 2017 to be consistent.

3.     Use of IRIS software for automatic coding and selection of the underlying cause of death 

a.     Reference 13 must be added at the end of the first paragraph or authors can add another reference concerning the IRIS software.

b.     A reference and a link to IRIS V5.8.0 must be added and the date of using it. 

c.     Also, a reference and link for the dictionary of medical terms used must be added.

d.     Same for ICD-10. 

4.     Statistical Analysis 

a.     A reference and a link to RevMan V.5 (Cochrane) must be added.

b.     A detailed description of the statistical model used to analyze the data must be added.

c.      More details for meta-analysis must be added.

d.     Did any regression analysis was performed in the meta-analysis? 

RESULTS

1.     Figure 1 must be moved to the Materials and Methods under the Population Study section as it presents the methodology subjects including criteria and the analysis workflow.

2.     Authors were using cervical cancer deaths and not uterus cancer they must change all the uterus cancer to cervical cancer over the manuscript.

1.     Excess deaths and excess death rates by cancer type 

a.     The authors are presenting a percentage of deaths, so they didn’t put the % after all numbers in this paragraph example “(36.2 deaths per 100,000 men)” is it 36.2% of 100,000 men. 

b.     There are non-definition of the excess deaths and the excess death rate in the methods section so how do the deaths are considered excess death did the authors use a specific range or scale above it considered an excess death?

c.     I think figure 4 is a table, not a figure. 

2.     Excess deaths and excess death rates by cancer type and age 

a.     Authors have stratified their cohort by gender and age they must devise their coherent in 3 groups according to cancer type in a subsequent paragraph if they want to get an idea about the death according to the cancer type and to reflect the title of the section.

3.     ORs for observed deaths vs. expected deaths by cancer type and age 

a.     In this section authors are not interpreting their ORs and CI results, they have shown an increase in all ORs that are >1 for the observed compared to expected but didn’t show the link with the fundamental results that were increased death due to the covid-19 pandemic and that are more risk for cancer patient’s survival.

4.     Excess deaths and excess death rates by cancer type and region 

a.     I would still prefer if the results will be presented in percentages.

b.     Authors must add the p values of each variable to the text and figures as they have mentioned in the methods section.

5.     English has to be improved.

DISCUSSION

1.     There are many updated studies concerning the pandemic effect on the health system that authors didn’t compare and discuss their results with the discussion that has to be refreshed and updated.

2.     There are many typos and spelling mistakes in the discussion section that must be corrected and improved.

CONCLUSION

Clear.

Author Response

Response-to-reviewers: Manuscript ID ijerph-2175734

 We thank the Reviewer for your comments and constructive criticism, we believe that the quality of our manuscript has been significantly improved. We have revised our paper in a point-by-point manner.

ABSTRACT.

Comment 1. The abstract missing the overall objective of the study’s authors didn’t clearly address the health issues caused by COVID-19 pandemic in the number of these cancers death and how they get the hypothesis of this study.

Response 1. Thank you for your comments. We have made the correction.

Comment 2. English must be improved.

Response 2. Thank you for your comments. We have made the correction.

INTRODUCTION

Comment 1. The first sentence of the manuscript “Burden disease and mortality varies by location and time.” Doesn’t have a scientific meaning authors must mention how and why because that’s due to multiple factors such as environment, lifestyle and genetics.

Response 1. Thank you for your comments. We have made the correction.

Comment 2. Authors missing a reference concerning “there were excess deaths of 18,2 million people worldwide”

Response 2. Thank you for your comments. We have made the correction.

Comment 3. This sentence is not clear and must be re-written “Therefore, COVID-19 infection was potentially a main cause of mortality. In addition to COVID-19 deaths, social distancing, and pandemic restrictions, societal and behavioral changes associated with the pandemic overstretched healthcare systems; other country-specific factors may have increased non- COVID-19 death rates, such as deaths from chronic diseases, including cancer [2]”

Response 3. Thank you for your comments. We have made the correction.

Comment 4. Authors need to present a clear link between the increase in deaths by Covid19 not just because of the infection but also due to lifestyle changes which may affect the screening programs, diagnostic delays, and treatment response to author diseases including cancers.

Response 4. Thank you for your comments. We have made the correction.

Comment 5. The English are too poor.

Response 5. Thank you for your comments. We have made the correction.

MATERIALS AND METHODS

  1. Design Study

Comment a. Usually, we say study design I don’t know if this title is suggested by the journal formatting.

Response a. Thank you for your comments. We have made the correction.

Comment b. Authors must clarify the design of their study in this section not why they chose to use the population-based electronic death 82 records in Peru.

Response b. Thank you for your comments. The study design is described in this section. We use SINADEF because it is the only system that records causes of death in Peru.

  1. Population Study

Comment a. Ethical approval and IRB registration number must be added to this section.

Response a. Thank you for your comments. We used publicly available death data of the SINADEF in Peru. These data are fully anonymous and are published as part of routine surveillance. Therefore, the study was exempt from review by an ethics board.

Comment b. Usually, we present the date by month, day than the year: the date “1 January 2017” must be presented as January 1st, 2017 to be consistent.

Response b. Thank you for your comments. We have made the correction.

  1. Use of IRIS software for automatic coding and selection of the underlying cause of death

Comment a. Reference 13 must be added at the end of the first paragraph or authors can add another reference concerning the IRIS software.

Response a. Thank you for your comments. We have made the correction.

Comment b. A reference and a link to IRIS V5.8.0 must be added and the date of using it.

Response b. Thank you for your comments. We have made the correction.

Comment c. Also, a reference and link for the dictionary of medical terms used must be added.

Response c. Thank you for your comments. We have made the correction.

Comment d. Same for ICD-10.

Response d. Thank you for your comments. We have made the correction.

  1. Statistical Analysis

Comment a. A reference and a link to RevMan V.5 (Cochrane) must be added.

Response a. Thank you for your comments. We have made the correction.

Comment b. A detailed description of the statistical model used to analyze the data must be added.

Response b. Thank you for your comments. Estimates of excess mortality (excess mortality in number, P-Score (%), and excess mortality rate (per 100,000 habitants)) are shown in lines XX-XX. There is no statistical program to analyze excess mortality.

Comment c. More details for meta-analysis must be added.

Response c. Thank you for your comments. We have made the correction.

Comment d. Did any regression analysis was performed in the meta-analysis?

Response d. Thank you for your comments. Adjusted analyzes cannot be performed due to the limited number of variables. In addition, adjusted analyzes cannot be performed in the RevMan software.

Results

Comment 1. Figure 1 must be moved to the Materials and Methods under the Population Study section as it presents the methodology subjects including criteria and the analysis workflow.

Response 1. Thank you for your comments. However, this cross-sectional, retrospective study following the Strengthening the Reporting of Observational Studies in Epidemiology reporting guidelines (STROBE; Appendix A. Table A1. Checklist STROBE). According to the Results section (STROBE; Appendix A. Table A1. Checklist STROBE). Participant. (c) Consider use of a flow diagram. Therefore, the "flow diagram" must be included in the Results section.

Comment 2. Authors were using cervical cancer deaths and not uterus cancer they must change all the uterus cancer to cervical cancer over the manuscript.

Response 1. Thank you for your comments. We select deaths from uterus cancer.

  1. Excess deaths and excess death rates by cancer type

Comment a. The authors are presenting a percentage of deaths, so they didn’t put the % after all numbers in this paragraph example “(36.2 deaths per 100,000 men)” is it 36.2% of 100,000 men.

Response a. Thank you for your comments. The sentence "In 2020, there was a total of 1729 prostate cancer deaths (36.2 deaths per 100,000 men), 1463 breast cancer deaths (19.8 deaths per 100,000 women), and 766 uterus cancer deaths (10.3 deaths per 100,000 women)" is the value observed in 2020. The percentage is for excess mortality with P-Score, not for the observed or expected value. Excess mortality also was calculated using the P indicator, which is the quotient of this difference between observed and expected deaths divided by expected deaths, and multiplied by 100 (expressed in percent) ((observed deaths−expected deaths)/expected deaths×100%).

Comment b. There are non-definition of the excess deaths and the excess death rate in the methods section so how do the deaths are considered excess death did the authors use a specific range or scale above it considered an excess death?

Response b. Thank you for your comments. However, excess deaths, expected deaths and observed deaths, according to the literature have been defined in section 2.5. Statistical Analysis "We estimated the excess deaths and the excess death rates due to cancer according to cancer type, age, and region. Expected cancer deaths in 2020 were obtained from the average of deaths in the 3 years preceding the pandemic (2017-2019). Cancer deaths (from prostate, breast, and uterine) reported from 1 January to 31 December 2020 were defined as observed deaths in 2020. Excess cancer deaths were defined as the difference between the number of observed cancer deaths in 2020 (during the COVID-19 pandemic) and the number of expected cancer deaths based on past deaths in the previous 3 years (average number of cancer deaths that occurred between 2017 and 2019) [2,14]. Excess mortality also was calculated using the P indicator [14], which is the quotient of this difference be-tween observed and expected deaths divided by expected deaths, and multiplied by 100 (expressed in percent) ((observed deaths−expected deaths)/expected deaths×100%) [14]. We calculated the cancer mortality rate observed and cancer mortality rate expected (per 100,000 habitants) by dividing the number of deaths per cancer type, region, and age by the estimated population of each region. The excess death rates by cancer type, age, and region per 100,000 population were calculated as difference between the mortality rate observed and the mortality rate expected [14]".

Comment c. I think figure 4 is a table, not a figure.

Response c. Thank you for your comments. Figure 4 is a figure. This Figure shows the trends in monthly excess deaths for prostate, breast, and uterus cancer between January and December 2020 in Peru. Color intensity is proportional to rank number (denoted by dark red for "highest excess" to "lowest excess" denoted by dark green).

  1. Excess deaths and excess death rates by cancer type and age

Comment a. Authors have stratified their cohort by gender and age they must devise their coherent in 3 groups according to cancer type in a subsequent paragraph if they want to get an idea about the death according to the cancer type and to reflect the title of the section.

Response a. Thank you for your comments. We have made the correction.

  1. ORs for observed deaths vs. expected deaths by cancer type and age

Comment a. In this section authors are not interpreting their ORs and CI results, they have shown an increase in all ORs that are >1 for the observed compared to expected but didn’t show the link with the fundamental results that were increased death due to the covid-19 pandemic and that are more risk for cancer patient’s survival.

Response a. Thank you for your comments. However, according to the IRIS analysis, deaths whose basic cause of death was prostate, breast and uterus cancer were selected. In SINADEF, data on co-infection with COVID-19 were not available to carry out survival analysis. Despite of this, we have included the following limitation "Another limitation was lack of data to confirm if the deaths were due to exacerbation of cancer symptoms or COVID-19 related comorbidity".

  1. Excess deaths and excess death rates by cancer type and region

Comment a. I would still prefer if the results will be presented in percentages.

Response a. Thank you for your comments. To present the homogeneous results, the "Excess deaths (n and %) and excess death rates (per 100,000) by cancer type and region" have been calculated. Therefore, we consider that this result is maintained in the current presentation.

Comment b. Authors must add the p values of each variable to the text and figures as they have mentioned in the methods section.

Response a. Thank you for your comments. P-values <0.05 was considered as significant only for meta-analytic methods. Excess mortality is not calculated with p-value.

  1. English has to be improved.

Response a. Thank you for your comments. We have made the correction.

DISCUSSION

Comment 1. There are many updated studies concerning the pandemic effect on the health system that authors didn’t compare and discuss their results with the discussion that has to be refreshed and updated.

Response 1. Thank you for your comments. We include a reference on excess mortality from cancer.

Comment 2. There are many typos and spelling mistakes in the discussion section that must be corrected and improved.

Response 2. Thank you for your comments. We have made the correction.

CONCLUSION

Comment 1. Clear.

Response 1. Thank you for your comments. 

Author Response

Response-to-reviewers: Manuscript ID ijerph-2175734

We thank the Reviewer for your comments and constructive criticism, we believe that the quality of our manuscript has been significantly improved. We have revised our paper in a point-by-point manner.

 Comment. The article titled “Estimating Excess Mortality due to Prostate, Breast, and Uterus Cancer during the COVID-19 Pandemic in Peru: A Retrospective, Cross-Sectional Study” by Ramírez Soto et. al. discusses the observed mortality in Peru during Covid-19 pandemic in 2020 due to deaths of patients with three different cancer types- Breast, Prostrate and Uterine Cancer. Such studies have been previously published by other nations – UK, USA, India etc., so the idea is not novel, neither are the findings – which show an increase in mortality in patients with different cancer types, as expected.

 Response. Thank you for your comments. Although other studies have been published in other regions, in Peru there is not enough information on excess mortality from cancer. There is only one study that has several limitations.

Comment 1. Line 72-74 is untrue. There is a published report that discusses the increase in cancer related mortality in Peru in 2020 due to Covid lockdowns, from March 15 to June 30. The article have also reported the increase in death due to Prostrate and Breast cancer, along with leukemia. While the authors here have shown their findings through trends, plots and tables and described the methods adequately, they have not cited the published study in this article anywhere, which is a must - Vidaurre et. al., Lancet Oncol. 2022 D.O.I. 10.1016/S0140-6736(21)02796-3.

 Response 1. Thank you for your comments. In the new version of the manuscript, we have cited the study Vidaurre et al. Excess mortality in patients with cancer during the COVID-19 pandemic in Peru: an analysis of death registry data. Lancet Oncol. 2022 Jul;23:S28.

Comment 2. Even though the authors have addressed the shortcomings of their study, the main issue is the lack of data to confirm if the deaths were due to exacerbation of cancer symptoms or covid-19 related comorbidity. This makes the reported % change less reliable.

Response 2. Thank you for your comments. However, according to the IRIS analysis, deaths whose basic cause of death was prostate, breast and uterus cancer were selected. In SINADEF, data on co-infection with COVID-19 were not available. Despite of this, we have included the following limitation "Another limitation was lack of data to confirm if the deaths were due to exacerbation of cancer symptoms or COVID-19 related comorbidity".

Comment 3. The authors have reported the statistical findings in the discussion but have not explained the significance of their findings.

Response 3. Thank you for your comments. We include a paragraph on implications for public health in Peru.

Round 2

Reviewer 2 Report

Authors have sufficiently addressed my concerns and comments.  

Author Response

Thank you